# Views Can Be Deceiving: Improved SSL Through Feature Space Augmentation

**Kimia Hamidieh**[1]    **Haoran Zhang**[1]    **Swami Sankaranarayanan**[2]    **Marzyeh Ghassemi**[1]
[1]MIT, [2]Sony AI
{hamidieh,haoranz,swamiviv,mghassem}@mit.edu

## Abstract

Supervised learning methods have been found to exhibit inductive biases favoring simpler features. When such features are spuriously correlated with the label, this can result in suboptimal performance on minority subgroups. Despite the growing popularity of methods which learn from *unlabeled* data, the extent to which these representations encode spurious features is unclear. In this work, we explore the impact of spurious features on Self-Supervised Learning (SSL) for visual representation learning. We first empirically show that commonly used augmentations in SSL can cause undesired invariances in the image space, and illustrate this with a simple example. We further show that classical approaches in combating spurious correlations, such as dataset re-sampling during SSL, do not consistently lead to invariant representations. Motivated by these findings, we propose LateTVG to remove spurious information from these representations during pretraining, by regularizing later layers of the encoder via pruning. We find that our method produces representations which outperform the baselines on several benchmarks, without the need for group or label information during SSL.

## 1 Introduction

Standard supervised machine learning models exhibit high overall performance but often perform poorly on minority subgroups (Shah et al., 2020; McCoy et al., 2019; Gururangan et al., 2018). One potential cause is the presence of spurious correlations, which are features that are only correlated with the label for specific subsets of data. For instance, a machine learning model tasked with predicting bird species from images across different habitats may use the background the bird commonly appears in as a "shortcut", instead of core features specific to the bird such as the shape of their beak or plumage. This results in poor performance on bird groups that appear in unexpected environments (Sagawa et al., 2020a). Identifying spurious correlations in the supervised learning setting has been well studied, where empirical risk minimization has been shown to exploit spurious correlations and result in poor performance for minority subgroups (Hashimoto et al., 2018). As downstream tasks are explicitly defined, the label can be used to distinguish between core and spurious features (Liu et al., 2021a; Zhang et al., 2022). Recent work has proposed various methods to identify and mitigate the effects of spurious features, such as learning multiple prediction heads (Lee et al., 2022b), causal inference (Creager et al., 2021), data augmentation (Gao et al., 2023) and targeted strategies such as importance weighting (Lahoti et al., 2020), re-sampling Idrissi et al. (2021); Tu et al. (2020), or approaches based on group distributionally robust optimization (Sagawa et al., 2020a; Duchi et al., 2019).

More recently, self-supervised learning (SSL) has emerged as a common form of pre-training for task-agnostic learning with large, unlabeled datasets (Chen et al., 2020a; He et al., 2019; Grill et al., 2020; Chen & He, 2020; Caron et al., 2020; Zbontar et al., 2021; Chen et al., 2020b). SSL methods learn representations from unlabeled datasets by solving an auxiliary pretext task (Doersch et al., 2015), such as inducing invariance between the representations of two augmented views of the same image (He et al., 2019; Chen et al., 2020a). These methods have shown impressive results for a wide range of downstream tasks and datasets (Liu et al., 2021b; Jaiswal et al., 2020; Tamkin et al., 2021).

Capturing *core* features – rather than spurious features – is essential for learning effective representations that can be used in downstream tasks, but is particularly difficult in the case of SSL due to the absence of labeled data during the pre-training process. Given only unlabeled data, we define spurious features as those that strongly correlate with core features for most examples in the training set, but

are not useful for downstream tasks. For example, when training an SSL model on multi-object images, larger objects may interfere with the learning of smaller objects (Chen et al., 2021). If the downstream task involves only the prediction of smaller objects, the larger (spurious) object may suppress the smaller (core) object from being learned. Large-scale unlabeled datasets that are commonly used in machine learning are inevitably imbalanced (Van Horn et al., 2021), have been found to be biased towards spuriously correlated sensitive attributes (Calude & Longo, 2017) such as gender or race (Agarwal et al., 2021), and can also include label-irrelevant features (Torralba & Efros, 2011; Fan et al., 2014).

In this paper, we investigate the impact of spurious correlations on SSL pre-training. We first show theoretically that image augmentations used in SSL pre-training can lead to spurious connectivity when learning representations, causing the model to fail to predict the label using core features in downstream tasks. We empirically evaluate spurious connectivity, and then show that existing methods for utilizing group information in ERM based approaches do not provide an analogous improvement in SSL pre-training. We then propose **Late**-*layer* **Transformation**-*based View Generation* or LATETVG – a method that induces invariance to spurious features in the representation space by regularizing final layers of the featurizer via pruning. Importantly, since our approach addresses SSL pre-training, we do not assume that model developers know *apriori* the identity or values of the spurious features that exist in the data. We first evaluate LATETVG on several popular benchmarks for spurious feature learning, and then connect our method to the theoretical analysis by showing that LATETVG models empirically exhibit lower spurious connectivity. Our method demonstrates improved discriminative ability, especially over minority subgroups, for downstream predictive tasks, without access to group or label information. We make the following contributions:

- We provide theoretical arguments (Sec 3.3) that illustrate how common augmentations used in SSL pre-training affect the model's ability to rely on spurious features, for downstream linear classifiers.
- We explore the extent of spurious learning in self-supervised representations through the lens of downstream worst-group performance. We empirically show that known techniques for avoiding spurious correlations, such as re-sampling of the training set given group information, do not consistently improve core feature representations (Sec 4.4).
- We propose LATETVG – an approach that corrects for the biases caused by augmentations, by modifying views of samples in the representation space (Sec 5.1). We find that LATETVG effectively improves worst-group performance in downstream tasks on four datasets by enforcing core feature learning (Sec 5.2).

## 2 RELATED WORK

**Spurious Correlations.** Spurious correlations arise in supervised learning models Koh et al. (2021); Joshi et al. (2023); Singla & Feizi (2021) in a variety of domains, from medical imaging (Zech et al., 2018; DeGrave et al., 2021) to natural language processing (Tu et al., 2020; Wang & Culotta, 2020). A variety of approaches have been proposed to learn classifiers which do not make use of spurious information. Methods like GroupDRO (Sagawa et al., 2020a) and DFR (Kirichenko et al., 2022) require group information during training, while methods like JTT (Liu et al., 2021a), LfF (Nam et al., 2020), CVaR DRO (Duchi et al., 2019), and CnC (Zhang et al., 2022) do not. However, all methods require group information for model selection.

**Self-supervised Representation Learning.** Self-supervised learning methods learn representations from large-scale unlabeled datasets where annotations are scarce. In vision applications, the pretext task is typically to maximize similarity between two augmented views of the same image (Jing & Tian, 2020). This can be done in a contrastive fashion using the InfoNCE loss (Oord et al., 2018), such as in Chen et al. (2020a) and Chen et al. (2020b), or without the need for negative samples at all, as in Grill et al. (2020); Caron et al. (2020); Chen & He (2020); Caron et al. (2021); Oquab et al. (2023); Zbontar et al. (2021). Prior work has shown that SSL models may learn to spuriously associate certain foreground items with certain backgrounds (Meehan et al., 2023), In this work, we explore one potential mechanism for this phenomenon, both theoretically and empirically.

**Representation Learning under Dataset Imbalance and Shortcuts.** Self-supervised models have demonstrated increased robustness to dataset imbalance (Liu et al., 2021b; Jiang et al., 2021b;a), and the dominance of easier or larger features suppressing the learning of other features (Chen et al., 2021). Some prior work has addressed shortcut learning in contrastive learning through adversarial feature modification without group labels (Robinson et al., 2021). However, other approaches to

group robustness or fairness in self-supervised learning require group information or labels (Tsai et al., 2020; Song et al., 2019; Wang et al., 2021; Bordes et al., 2023; Scalbert et al., 2023). This paper focuses on learning representations from an unlabeled dataset with spurious correlations, encompassing both dataset imbalance and features of varying difficulty.

**Regularization in Self-supervised Learning.** The concept of regularizing a specific subset of the network is relatively unexplored in self-supervised learning but finds motivation in recent findings from supervised settings, such as addressing minority examples (Hooker et al., 2019), out-of-distribution generalization (Zhang et al., 2021), late-layer regularizations through head weight-decay (Abnar et al., 2021), and initialization (Zhou et al., 2022). Additionally, Lee et al. (2022a) propose surgically fine-tuning specific layers of the network to handle distribution shifts in particular categories. These studies provide support for the approach of targeting a specific component of the network in self-supervised learning.

## 3 SPURIOUS CONNECTIVITY INDUCES DOWNSTREAM FAILURES

In this section, we introduce a toy setting to demonstrate that common augmentations used in SSL pre-training affect a model's ability to rely on spurious features for downstream linear classifiers. We consider a binary classification problem with a binary spurious attribute, with an equal number of samples per group (Section 3.2). We show that augmentations applied during SSL pre-training can introduce undesired invariances in the representation space learned by a contrastive objective, making the downstream linear classifier trained on representations more reliant on the spurious feature (Section 3.3).

### 3.1 BACKGROUND AND SETUP

**Setup.** BWe consider learning representations from an unlabeled data space $\mathcal{X}$ generated from an underlying latent feature space $\mathcal{Z} \in \mathbb{R}^m := \{z_{\text{core}}, z_{\text{spur}}, \ldots, z_m\}$, where $z_{\text{core}}$ and $z_{\text{spur}}$ are correlated features. For a given downstream task with labeled samples, we assume that each $x \in \mathcal{X}$ belongs to a class given by the ground-truth labeling function $y : \mathcal{X} \to \mathcal{Y}$ where $z_{\text{core}}$ determines the labels for our downstream task of interest, while $z_{\text{spur}}$ determines the spurious attribute, which is easier to learn, and is not of interest for downstream tasks. We can define a deterministic attribute function $a : \mathcal{X} \to \mathcal{S}$ where each $x \in \mathcal{X}$ takes a value in $\mathcal{S}$. Let $g = (y(x), a(x))$ denote the subgroup of a given sample $x$, where $\mathcal{G} = \mathcal{Y} \times \mathcal{S}$ is the set of all possible subgroups. Figure 1 illustrates the subgroups on the Waterbirds dataset, where the background is a spurious feature that correlates with the bird species.

**Contrastive learning.** We aim to learn representations by bringing together data-augmented views of the same input, which we refer to as positive pairs, using a contrastive objective. Let $P_+$ be the distribution of positive pairs, which can be defined as the marginal probability of generating the augmented pair $x$ and $x'$ from the same image in the (natural) population data. Thus the distribution $P_+$ relies both on original data distribution and the choice of SSL augmentations. To analyze the representation space learned in contrastive learning and core feature predictivity of the representations, consider a weighted graph with vertex set $\mathcal{X}$ where the undirected edge $(x, x')$ has weight $w_{xx'} = P_+(x, x')$ similar to augmentation graph in HaoChen et al. (2021).

Although the augmentation graph learns semantically similar structures that enables generalization to new domains (Shen et al., 2022), the inductive biases set by these augmentations is not well studied. In this work, we draw attention to cases where augmentations can create *spurious connectivities* within subgroups of the data, and when and why these connectivities can cause the downstream linear model to rely on the spurious feature.

### 3.2 SPURIOUS CONNECTIVITY IN A TOY SETUP

In this section, we introduce a setting in which contrastive objectives can learn representations that cause linear downstream models fail on downstream tasks. To start, we investigate how augmentations can transform the samples such that the subgroup assignment changes.

**Definition 3.1.** *Subgroup connectivity.* Define the average subgroup connectivity given two disjoint subsets $G_1, G_2 \subseteq \mathcal{X}$ as $w(G_1, G_2) = \frac{1}{|G_1|.|G_2|} \sum_{x \in G_1, x' \in G_2} w_{xx'}$. where $w_{xx'}$ is the probability of generating the augmented pair $x$ and $x'$ from the same image in the natural population data.

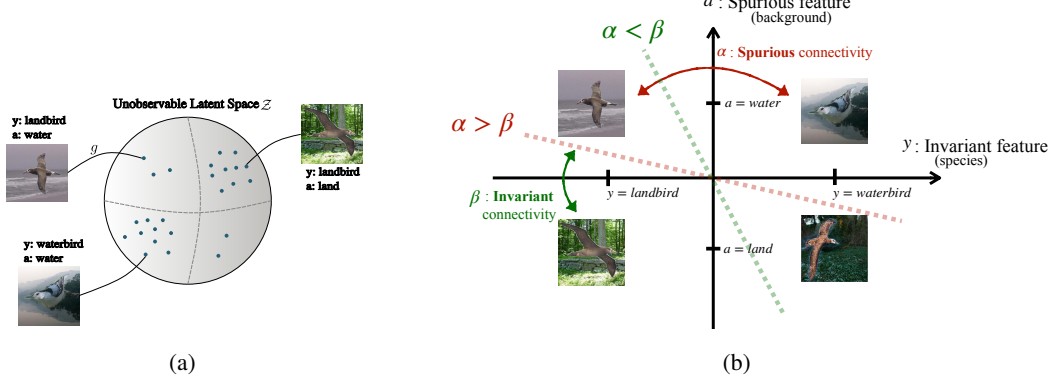

Figure 1: **Analysing SSL augmentations**. (a) Images generated from a latent space with correlating features. (b) If the connectivity induced by SSL augmentations between subgroups with the same spurious features is higher than the ones with the same invariant features, learned representations lead a downstream linear model to separate the data based on the spurious feature (red dashed line) instead of the invariant feature (green dashed line). Our empirical evaluation in Table 4 shows that this is indeed the case across different datasets considered in this work.

Intuitively, this subgroup connectivity is the average weight of edges connecting $G_1$ to $G_2$, and is proportional to the probability of a sample $x \in G_1$ being transformed to a sample $x' \in G_2$ via augmentations. See Appendix C for further details.

We specifically define the following terms to be the expected value of $w(G_1, G_2)$ from Definition 3.1, when subgroups $G_1$ and $G_2$ have the following properties:

- **Spurious connectivity** ($\alpha$): $G_1$ and $G_2$ share the same spurious attribute but differ in class
- **Invariant connectivity** ($\beta$): $G_1$ and $G_2$ share the same class but differ in spurious attribute
- **Opposite connectivity** ($\gamma$): $G_1$ and $G_2$ differ both in the spurious attribute and the label

Where $\alpha$, $\beta$, $\gamma$ are average values estimated across a dataset consisting of subgroups.

**Toy Setup.** We consider a downstream classification problem where a spurious attribute is present, and both the input and the spurious attribute take binary values. We define the probability of sampling a positive pair $(x, x')$ based on the expected connectivity terms $\alpha_{\text{toy}}$, $\beta_{\text{toy}}$, $\gamma_{\text{toy}}$, and $\rho_{\text{toy}}$ as follows:

$$P_+(x, x') = \begin{cases} \alpha_{\text{toy}}, & \text{if } a(x) \neq a(x') \text{ and } y(x) = y(x') \\ \beta_{\text{toy}}, & \text{if } a(x) = a(x') \text{ and } y(x) \neq y(x') \\ \gamma_{\text{toy}}, & \text{if } a(x) \neq a(x') \text{ and } y(x) \neq y(x') \\ \rho_{\text{toy}}, & \text{if } a(x) = a(x') \text{ and } y(x) = y(x') \end{cases}$$

Note that the average subgroup connectivity for this setup, would be exactly the same as the corresponding connectivity variable. Thus in our running example we have $\alpha = \alpha_{\text{toy}}, \beta = \beta_{\text{toy}}, \gamma = \gamma_{\text{toy}}$, and we can use them interchangeably. For this simplified augmentation graph, the expected connectivity terms between groups are a property of the graph, and independent of the model or architecture we use for learning representations. Combined with a contrastive objective, the expected connectivity can be a proxy for how close different subgroups are going to be in the representation space.

### 3.3 ANALYSIS OF THE TOY SETTING

In Section 4.2, we empirically show that common augmentations used in contrastive learning can be detrimental to learning invariant representations, as they implicitly encourage samples to cluster primarily based on the spurious feature. Based on this observation, we make the following assumption.

**Assumption 3.2.** Given a spurious attribute function $a : \mathcal{X} \to |G|$ which is defined for all $x \in \mathcal{X}$, we assume that for a data point $x \in \mathcal{X}$, the probability of distorting the labeling of the augmented images sampled from the augmentation distribution $\mathcal{A}(\cdot|\bar{x})$, is greater than the probability of distorting the attribute. More formally,

$$\Pr_{\tilde{x} \sim \mathcal{A}(\cdot|x)} (y(\tilde{x}) \neq y(x), a(\tilde{x}) = a(x)) \geq \Pr_{\tilde{x} \sim \mathcal{A}(\cdot|x)} (y(\tilde{x}) = y(x), a(\tilde{x}) \neq a(x))$$

**Lemma 3.3.** *Consider the set of (unlabeled) population data $\mathcal{X}$ in a binary-class setting where the spurious attribute takes binary values, consisting of $|\mathcal{G}| = 4$ groups, with the same number of*

*examples per group. Consider a simplified augmentation graph with parameters $\alpha$, $\beta$, $\rho$, $\gamma$ defined as in 3.2, and assume that augmentations are more likely to change either class or attribute, than to change neither of the two ($\alpha > \gamma, \beta > \gamma$), and that augmentations are less likely to change both at the same time ($\rho > \alpha, \rho > \beta$).*

*Under these conditions, the spectral contrastive loss recovers both invariant and spurious features, and for each sample in the population data, the spurious feature is bounded by constant $B_{sp} = \sqrt{\beta - \alpha - \gamma + \rho}$, while the invariant feature is bounded by $B_{inv} = \sqrt{\alpha - \beta - \gamma + \rho}$, in the representation space. Proof in Appendix C.*

**Corollary 3.4.** *Given Assumption 3.2, where $\alpha > \beta$ in the simplified augmentation graph, the margin of the spurious classifier is $B_{sp}$, and is less than the margin of the invariant classifier $B_{inv}$, and the max-margin classifier trained on representations given by spectral clustering converges to the spurious classifier.*

This suggests that even with the same number of samples across different groups during pre-training, downstream linear classifiers can rely on the spurious feature to make predictions, where the representations are determined by the simplified augmentation graph and the spectral contrastive loss.

## 4 EXPLORING SPURIOUS LEARNING IN REPRESENTATIONS

In this section, we investigate the performance of downstream linear models trained on self-supervised representations, empirically verify our assumption regarding spurious and invariant connectivity, and show that in practice – similar to our toy analysis – having the same number of examples across groups in the presence of spurious connectivity does not lead to performance gains.

### 4.1 EXPERIMENTAL SETUP

**Datasets** We evaluate methods on five commonly used benchmarks in spurious correlations – CelebA (Liu et al., 2015), CMNIST (Arjovsky et al., 2019), MetaShift (Liang & Zou, 2022), Spurious CIFAR-10 (Nagarajan et al., 2020), and Waterbirds (Wah et al., 2011) (See Appendix D.1 for dataset descriptions). For each dataset, we train an encoder with an SSL-based pre-training step followed by a supervised training of a linear model that probes the representations learned using SSL for the downstream task.

**SSL Pre-training** For the SSL pre-training, we train SimSiam (Chen & He, 2020) models with a ResNet backbone throughout the paper. The training split used during the pre-training stage are unbalanced and contain spuriously correlated data. The group/label counts for each dataset and split is shown in Appendix D.1. The backbone network used for most of our experiments are initialized with random weights, unless specified otherwise. We additionally report results for SimCLR (Chen et al., 2020a) models in Section 5.2.1.

**Downstream Task** For downstream task prediction, we train a linear layer using logistic regression on top of the pretrained embeddings. Note that the backbone is frozen during this finetuning phase and only the linear layer is updated. We use a balanced dataset for training where the spurious correlation does not hold. To create this downstream training dataset, we subsample majority groups (Sagawa et al., 2020b; Idrissi et al., 2021), to avoid the geometrical skews (Nagarajan et al., 2020) of the linear classifier on representations. Then, we evaluate the learned representations on the standard test split of each dataset, where group information is given. For each run, we report the average and worst-group accuracy.

**Empirical Evaluation of Spurious Connectivity** To evaluate the connectivity term for each pair of subgroups in datasets exhibiting spurious correlations, we conduct an empirical analysis similar to Shen et al. (2022). Specifically, we train a classifier to distinguish between each pair of subgroups and evaluate its performance on a subset of the data that has been augmented with SSL augmentations. The error of the classifier represents the probability that the augmentation module alters the subgroup assignment for each example between the two subgroups, making them indistinguishable. Figure 1 illustrates this procedure.

**The Role of Initialization** In representation learning, encoders are not typically trained from scratch but initialized from a model pretrained on larger datasets, such as ImageNet (Deng et al., 2009). Recent work in transfer learning (Geirhos et al., 2018; Salman et al., 2022) has questioned this assumption and pointed out that biases in pretrained models linger even after finetuning on downstream target tasks. In this section and more broadly in our work, we focus on performing SSL pre-training from randomly initialized weights. In addition, since the datasets considered in this work

are similar to ImageNet, the performance of off-the-shelf ImageNet pretrained models is expected to be higher. For completeness, we have added these results to Appendix G.2.

## 4.2 HIGH LEVELS OF SPURIOUS CONNECTIVITY IN PRACTICE

We measure connectivity across four datasets in Table 4, and on all of them, we find that the average spurious connectivity is higher than invariant connectivity. We also confirm that both these values are higher than the probability of simultaneously changing both spurious attributes and invariant attributes. This means that the samples within the training set are more likely to be connected to each other *through* the spurious attribute, rather than the core feature. This suggests that the contrastive loss prefers alignment based on the spurious attribute instead of the class.

Table 1: We report the error of classifiers trained to distinguish between two subgroups as a proxy for the probability of augmentations flipping group assignments between each two groups in the dataset, or the connectivity of two subgroups in the image space.

| Dataset | Spurious Connectivity | Invariant Connectivity | Opposite Connectivity |
|---|---|---|---|
| celebA | 10.4 | 3.7 | 2.8 |
| cmnist | 31.6 | 8.3 | 6.8 |
| metashift | 16.3 | 13.6 | 5.0 |
| waterbirds | 25.3 | 11.2 | 7.8 |

We compute the connectivity terms by training classifiers to distinguish augmented data from each combination of the two groups in the dataset and reporting their error rates.

The details of the choice of augmentations and training for this step can be found in Appendix E.

## 4.3 SSL MODELS LEARN SPURIOUS FEATURES

To measure the reliance of downstream models to spurious correlations, we measure the accuracy of the downstream model on each group in the test set, and use the worst-performing group accuracy as a lens to reason about spurious correlations. We find across all datasets, SSL models exhibit gaps between worst-group and average accuracy when predicting the core feature (Table 5 in Appendix D.3).

These results indicate, that unlike supervised learning (Menon et al., 2021; Kirichenko et al., 2022; Rosenfeld et al., 2022), training of the final layer on a balanced set where the spurious correlation does not hold is not sufficient for improving worst-group accuracy when predicting the core attribute.

## 4.4 RESAMPLING DURING SSL DOES NOT IMPROVE DOWNSTREAM PERFORMANCE

To probe the effect of availability of such group information during the SSL pre-training stage, we examine whether classical approaches for combating spurious correlations, such as re-sampling training examples (Idrissi et al., 2021), are effective in removing spurious information during SSL pre-training.

Assuming that group information is available, we train SimSiam on datasets re-sampled using the following strategies: (i) *Balancing* groups by resampling training examples to match the downstream validation distribution. (ii) *Downsampling* examples in majority groups to have the same number of examples in all groups. (iii) *Upsampling* minority examples to have the same number of examples in all groups.

Table 2: **Worst-group accuracy difference** (%) between each balancing strategy and the original training set. Original training performance are shown in parentheses below each dataset. Full results can be found in Appendix Table 10.

| Sampling Strategy | celebA (77.5) | cmnist (75.4) | metashift (42.3) | spurcifar10 (43.4) | waterbirds (48.3) |
|---|---|---|---|---|---|
| **Balancing** | -1.7 | -8.7 | -3.8 | -8.3 | +3.0 |
| **Downsampling** | +0.3 | -10.6 | +3.9 | -14.4 | +0.5 |
| **Upsampling** | +4.1 | -5.3 | +2.7 | -19.4 | -0.3 |

We find that re-sampling during self-supervised pre-training does not improve downstream worst-group accuracy in a consistent manner as in Table 2. We do see minor improvements for metashift

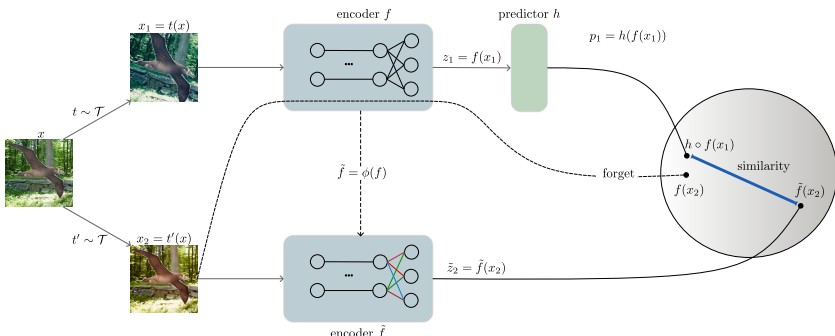

Figure 2: We use model transformation modules to create new views of training examples in the representation space. The introduced set of transformations removes the features learned in the final few layers, and provides final representations invariant to such transformations.

and `celebA`, but contrast this with large drops for `spurcifar10` and `cmnist`. Given that the downstream linear model is trained on a downsampled dataset where such correlations do not exist, this means that re-sampling during self-supervised training does not necessarily improve the linear separability of representations with respect to the core feature, even given a balanced finetuning dataset. This is analogous to our findings in the toy setting in Section 3.3.

## 5 CREATING ROBUST REPRESENTATIONS VIA FEATURE SPACE AUGMENTATIONS

In the previous sections, we showed that augmentation mechanisms used in SSL result in poor performance under spuriously correlated features in the training set. Instead of curating specific image augmentations that correct for these biases in the image space, we propose an approach to target spurious connectivity in the *representation* space by modifying positive pairs. In this section, we describe our approach, LATETVG that improves the performance of SSL models by introducing pruning based regularization to the later layers of the encoder.

### 5.1 LATE-LAYER TRANSFORMATION-BASED VIEW GENERATION

Motivated by improved SSL model invariance when trained with augmentations in *image* space (Chen et al., 2020a), we propose a model transformation module that specifically targets augmentations that modify the spurious feature in *representation* space. We propose *Late-layer Transformation-based View Generation* - LATETVG , which uses feature space transformations to mitigate spurious learning in SSL models and improve learning of the core feature.

Formally, we propose using a model transformation module $\mathcal{U}$, that transforms any given model $f_\theta$ parameterized by $\theta = \{W_1, \ldots, W_n\}$ to $f_{\tilde{\theta}}$. At each step, we draw a transformation $\phi_{M,\theta} \sim \mathcal{U}$ to obtain the transformed encoder. Each model transformation can be defined with a mask $M \in \{0, 1\}^{|\theta|}$, where we transform the unmasked weights $(1 - M) \odot \theta$ by $\phi$, and keep the rest of the weights $M \odot \theta$ the same to obtain $\tilde{\theta}$. Here, we propose a specific transformation module $\mathcal{U}$.

**Transformations.** For mitigating spurious connectivity, we choose a simple transformation targeted towards regularizing the final layers of the encoder. In our experiments, we consider a threshold pruning transformation module, which uses magnitude pruning on $a\%$ of the weights in all layers deeper than $L$. More specifically, we propose a model transformation module $\mathcal{U}_{\text{Prune, L, a}}$, with $\phi(\theta) = 0$, $M := M_{L,a} = \{M_L^l \odot \text{Top}_a(W_l) \mid l \in [n]\}$ and $\text{Top}_a(W_l)_{i,j} = \mathbb{I}(|W_{l_{(i,j)}}| \text{ in top } a\% \text{ of } \theta)$. Note that in this specific setting, the module transformation is deterministic (i.e. $|\mathcal{U}| = 1$), though our formalization also allows for random transformations such as randomized pruning or re-initialization.

To learn these representations, given two random augmentations $t, t' \sim \mathcal{T}$ from the augmentation module $\mathcal{T}$, two views $x_1 = t(x)$ and $x_2 = t'(x)$ are generated from an input image $x$. At each step, given a feature encoder $f$, and an augmentation module $\mathcal{U}$, we obtain a transformed model $\tilde{f} = \phi(f)$ with $\phi \sim \mathcal{U}$. During training, examples $x_1$ and $x_2$ are respectively passed through the normal encoder $v_1 = f(x_1)$, and the transformed encoder $\tilde{v}_2 = \tilde{f}(x_2)$. Encoded feature $\tilde{v}_2$ is now a positive example that should be close to $v_1$ in the representation space. An algorithmic representation of the method can be found in Appendix B.

Table 3: Worst-group accuracy (%) of SSL-Base and LATETVG for SimSiam and SimCLR pre-training. Results for average accuracy can be found in Table 8.

|  | **SimSiam** | | **SimCLR** | |
|---|---|---|---|---|
|  | SSL-BASE | SSL-LATE-TVG | SSL-BASE | SSL-LATE-TVG |
| celebA | 77.5 | **83.1** | 76.7 | **82.2** |
| cmnist | 80.7 | **83.1** | 81.7 | **83.8** |
| metashift | 42.3 | **79.6** | 45.5 | **59.3** |
| spurcifar10 | 43.4 | **61.4** | 36.5 | **40.4** |
| waterbirds | 48.3 | **56.3** | 43.8 | **55.4** |

**Intuition for LateTVG.** When learning a discriminative process that maps data to a separable space, the variance among different subpopulations is stored in distinct regions of the network (Lee et al., 2022a). As a result, both spurious and core features, which describe the high-level data distribution, tend to reside at the end of a neural network. Thus, in LATETVG , we aim to encourage the final layers to learn more difficult features, by applying a model transformation that targets these layers, and causing the model to be invariant to final layer transformations. As pruning in supervised models have been shown to affect minority examples more than majority ones (Hooker et al., 2019), we hypothesize that our transformation can be considered as a curated view generating operation for the minority groups. In particular, pruning would contribute to "forgetting" the minority examples from the network, resulting in upweighting the loss for these examples.

## 5.2 EXPERIMENTS

In this section, we demonstrate the efficacy of LATETVG in mitigating the dependence on spurious correlations. We use the same experimental setup as described in Section 4.1. For evaluation of LATETVG , we use our SSL-LATETVG approach during the pre-training stage. We compare this performance to SSL models pre-trained with the standard SSL-base trained with either SimSiam or SimCLR.

### 5.2.1 LATETVG IMPROVES SSL WORST-GROUP PERFORMANCE

The goal of this experiment is to understand how LATETVG affects worst-group performance in downstream tasks that use SSL representations. We compare the worst group accuracy of two approaches, SSL-Base and SSL-LATETVG on 5 different datasets. Both models used similar hyper-parameter grids and model selection criteria as noted previously. The results are presented in Table 3. We show the performance of the best hyperparameter combination here, and have provided figures of performance gains for all hyperparameters in Appendix D.2. It can be clearly observed that SSL-LATETVG outperforms the base model by large margins across most datasets and for both SimSiam and SimCLR. On cmnist, our performance is very close to the baseline model and we do not see significant improvement. We hypothesize that this is due to the fact that the base encoder on the easier cmnist dataset is already quite performant. On datasets where the base encoder performs poorly such as metashift and spurcifar10, our approach improves the performance by at least 10% over base SimSiam. On a dataset of a larger scale like celebA, LATETVG still improves upon a strong encoder baseline.

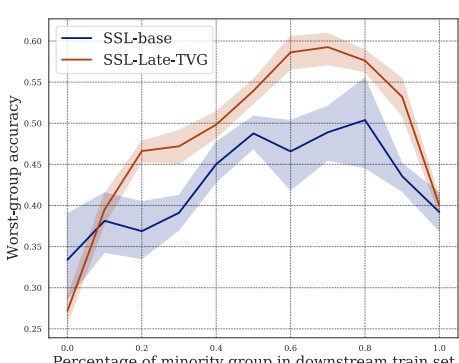

Figure 3: Downstream worst-group accuracy of SSL-Late-TVG on the metashift dataset as we vary the percentage of minority group in the downstream training set. For all cases except for extreme minority decrement, SSL-Late-TVG outperforms the baseline.

Further, we find that LATETVG closes the gap in performance to supervised pretraining (Table 8). We emphasize that this is an unfair comparison to begin with, since supervised pretraining requires labeled data whereas SSL does not, hence reducing the annotation budget drastically. Regardless, we find that LATETVG narrows the gap between the SSL baseline and the ERM model significantly

– 17% relative improvement for `cmnist` to 50% in the case of `spurcifar10`. In the case of `celebA`, we even outperform the ERM baseline.

### 5.2.2 SSL DOWNSTREAM LINEAR PERFORMANCE IS LESS RELIANT ON A BALANCED DOWNSTREAM DATASET

Traditional approaches that mitigate spurious correlations in ERM-based settings assume that the downstream training set is balanced (Kirichenko et al., 2022). However, this still requires knowledge of the spurious feature, which we may not always have in practice. In this experiment, we challenge this assumption and analyze how SSL models behave when the downstream training set is imbalanced.

We vary the proportion of minority groups in the downstream training set, by first downsampling the training set to have the same number of samples across groups, and second randomly sampling minority groups with weight $\lambda$ (x-axis in Figure 3) and majority groups with weights $1 - \lambda$. We measure the worst group accuracy of the trained linear models for each dataset. We show the results on `metashift` in Figure 3, comparing the performance of SSL-Base and SSL-LATETVG . We can observe that LATETVG outperforms the baseline across a range of minority weights – implying that LATETVG is more robust to imbalances in downstream training data. This is a crucial aspect where LATETVG differs from other approaches in the supervised pretraining literature, such as DFR (Kirichenko et al., 2022), which requires a balanced training set for the reweighting strategy to be successful. Similar results for other datasets and linear models are provided in in Appendix F.5.

### 5.2.3 LATETVG REDUCES SPURIOUS CONNECTIVITY IN THE REPRESENTATION SPACE

Finally, we relate our method back to the theoretical analysis presented in Section 3, by computing the connectivity of the *representation space* learned by the SSL models, using the procedure outlined in Section E. In Table 4, we find that LATETVG empirically reduces the spurious connectivity, while increasing the invariant connectivity, for all datasets. Thus, we have shown that LATETVG successfully augments the representation space to induce desired invariances.

Table 4: We report the error of classifiers trained to distinguish between the *representations* of two subgroups as a proxy for connectivity terms. We find that LATETVG decreases spurious connectivity while increasing invariant connectivity in comparison to the baseline.

| Dataset | Representation Space | Spurious Connectivity | Invariant Connectivity | Opposite Connectivity |
|---------|----------------------|-----------------------|------------------------|-----------------------|
| `celebA` | SSL-BASE | 18.9 | 15.7 | 8.3 |
| | SSL-LATE-TVG | 15.8 | 17.9 | 8.0 |
| `cmnist` | SSL-BASE | 37.3 | 3.2 | 2.7 |
| | SSL-LATE-TVG | 34.8 | 3.8 | 3.0 |
| `metashift` | SSL-BASE | 28.6 | 21.4 | 21.8 |
| | SSL-LATE-TVG | 27.3 | 27.3 | 21.3 |
| `waterbirds` | SSL-BASE | 44.9 | 9.4 | 8.4 |
| | SSL-LATE-TVG | 44.6 | 13.5 | 12.8 |

## 6 CONCLUSION

In this paper, we have investigated the impact of spurious correlations on self-supervised learning (SSL) pre-training and proposed a new approach, called LATETVG to address the issue. Our experiments demonstrated that spurious correlations caused by data augmentation can lead to spurious connectivity and hinder the model's ability to learn core features, which ultimately impacts downstream task performance. We have shown that traditional debiasing techniques, such as re-sampling, are not effective in mitigating the impact of spurious correlations in SSL pre-training. In contrast, LATETVG effectively improves the worst-group performance in downstream tasks by inducing invariance to spurious features in the representation space throughout training. Our approach does not require access to group or label information during training and can be applied to large-scale, imbalanced datasets with spurious correlations. We believe our work will help advance the field of SSL pre-training and encourage future research in developing methods that are robust to spurious correlations.

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
