# OpenReview forum: "Views Can Be Deceiving: Improved SSL Through Feature Space Augmentation"
_ICLR.cc/2024/Conference — ICLR 2024 spotlight_

### Official Review · Reviewer_32bS · 2023-10-29

**Soundness:** 2 fair
**Presentation:** 2 fair
**Contribution:** 2 fair
**Rating:** 5
**Confidence:** 4

**Summary:**

In this paper, the authors explore the self-supervised learning (SSL) in visual representation learning. They emphasize the impact of spurious features on model efficacy. Spurious features are those that are only correlated with the label for specific subsets of data, potentially leading to suboptimal results, especially for minority subgroups. This study delves into the extent to which SSL representations depend on such spurious features and introduces an innovative approach known as LATETVG to alleviate their impact during the pretraining phase.

**Strengths:**

* The introduced method effectively reduces spurious correlations and enhances the performance of downstream tasks across multiple datasets.

* The author presents a clear depiction of the prevalent issues with SSL arising from spurious features.

**Weaknesses:**

* The paper's structure and content are not easy to follow. Specifically, the use of symbols like alpha, beta, and gamma appears to be overloaded, making it challenging to distinguish between the connectivity and the P matrix.
* It's unclear how the correlation in Table 1 is computed. It seems spurious attributes and core features are not defined for real data.
* The theoretical foundation leans heavily on Spectral Contrastive Learning, yet it is not used in the experimental section. It begs the question of its relevance or applicability. And whether the empirical results are well justified by theory.
* The presentation would benefit significantly from the inclusion of an algorithmic representation of the proposed Late TVG method, enhancing its comprehensibility for readers.

**Questions:**

* Which training parameters were utilized in the LaterTVG method?
* How do the authors identify spurious features? The paper does not provide clear definitions distinguishing between spurious attributes and core features.

**Details Of Ethics Concerns:**

None.

---

> ### Author Response · Authors · 2023-11-17
> **Response to Reviewer 32bS**
>
> We thank the reviewer for their constructive feedback on our work. We appreciate the time taken to provide thoughtful comments to help strengthen our paper. We respond to each point below.
>
>
> > W1. The paper's structure and content are not easy to follow. Specifically, the use of symbols like alpha, beta, and gamma appears to be overloaded, making it challenging to distinguish between the connectivity and the P matrix.
>
> Thank you for pointing out the notation overload. To address this issue, we keep the symbol $\alpha$ at section 3.2 to mean the expected spurious connectivity across dataset, and have revised the notation when introducing the toy setup to clarify the relation between $\alpha_{\text{toy}}$, and $\alpha$. We agree that overload of these symbols made parts of the theory section difficult to follow, and have clarified this in the revised draft. If you have any additional suggestions for paper structure, please let us know.
>
>
> > W2. It's unclear how the correlation in Table 1 is computed. It seems spurious attributes and core features are not defined for real data.
>
> We originally discussed how the connectivity error rates in Table 1 are computed in Appendix E. We have added a short summary of this procedure to Section 4.2. To summarize, we utilize the spurious attributes given as metadata in each of the datasets, along with the label (i.e. the core attribute), to partition the dataset into different groups. We then compute the connectivity terms by training classifiers to distinguish augmented data from each combination of the two groups and reporting their error rates.
>
>
> > W3. The theoretical foundation leans heavily on Spectral Contrastive Learning, yet it is not used in the experimental section. It begs the question of its relevance or applicability. And whether the empirical results are well justified by theory.
>
> HaoChen et al. (2021) [1] shows that spectral contrastive learning achieves similar empirical performance to other contrastive learning methods, but enables easier theoretical analysis. Theoretically, the spectral contrastive loss is similar to many popular contrastive losses. For instance, the contrastive loss in SimCLR can be re-written as an expression similar to the spectral contrastive loss. We have added a short clarification to Appendix C.
>
>
>
> > W4. The presentation would benefit significantly from the inclusion of an algorithmic representation of the proposed Late TVG method, enhancing its comprehensibility for readers.
>
> Per the reviewer's suggestion, we have included the Late-TVG algorithm box including details in Appendix B to improve comprehension of the proposed method.
>
>
> > Q1. Which training parameters were utilized in the LateTVG method?
>
> The training parameters used for Late-TVG are identical to the SSL-Base model. The Late-TVG encoder parameters are modified via pruning during training. We have clarified this in Appendix D.2.
>
>
> > Q2. How do the authors identify spurious features? The paper does not provide clear definitions distinguishing between spurious attributes and core features.
>
> The spurious features are provided as metadata in each dataset we use, thus are known apriori (e.g. background in Waterbirds). We do not attempt to discover unknown spurious correlations. We have added text to clarify this assumption in Appendix D.1.
>
> We believe these changes address the major points raised by the reviewer. Please let us know if any part of our response remains unclear, or if you have any other suggestions to further improve the paper. We greatly appreciate you taking the time to provide feedback on our work.
>
>
> [1] HaoChen, Jeff Z., et al. "Provable guarantees for self-supervised deep learning with spectral contrastive loss." Advances in Neural Information Processing Systems 34 (2021): 5000-5011.

---

> > ### Author Response · Authors · 2023-11-21
> >
> > Dear Reviewer 32bS,
> >
> > Thank you again for your valuable feedback. Since the author response period is ending in less than two days, we were wondering if our response has adequately addressed your concerns. If so, we would appreciate it if you could update your review and raise your score accordingly. To briefly summarize, we have clarified notation, and provided additional methodological details in the appendices per your suggestion. If there are any remaining questions or comments, please let us know and we would be happy to discuss.
> >
> > Thank you!

---

### Official Review · Reviewer_BzfQ · 2023-11-01

**Soundness:** 4 excellent
**Presentation:** 4 excellent
**Contribution:** 3 good
**Rating:** 8
**Confidence:** 4

**Summary:**

This paper studies the problem of spurious correlations in self-supervised learning, in particular, contrastive learning. The paper first formally shows, in a toy setting, that ensuring the groups are balanced is not sufficient to ensure a spurious correlation is not learnt. This is then confirmed empirically as well. The paper then presents the new method to remedy this: "LATE-LAYER TRANSFORMATION-BASED VIEW GENERATION". This involves conducting magnitude pruning for the later layers of the network to remove dependence on spurious. This method is able to improve worst-group accuracy on some popular datasets to study spurious correlations in supervised learning.

**Strengths:**

1. The problem of spurious correlations has not been studied for SSL before and is of importance, considering the lack of group / label information in SSL can make this hard to remedy.

2. The conclusion that balancing the data may not be effective for remedying spurious correlations is extremely interesting.

3. The method proposed is effective in remedying worst-group accuracy despite other standard methods like group balancing failing.

**Weaknesses:**

1. The intuition / reasoning behind the choice to prune the later layers to forget the spurious feature is not well-explained / discussed sufficiently.

**Questions:**

1. The problem of spurious correlations in SSL is actually equivalent to **feature suppression** studied first here: https://proceedings.neurips.cc/paper/2021/hash/628f16b29939d1b060af49f66ae0f7f8-Abstract.html. It will be useful to discuss the equivalence of this problem as well as other relevant literature.

2. In supervised learning, the models trained on Waterbirds etc. are initialized from ImageNet pretrained weights. Is that the case here as well?

3. Experiments on some larger scale datasets provided in packages such as WILDS (https://wilds.stanford.edu/) or SpuCo (https://spuco.readthedocs.io/en/latest/) can strengthen the empirical success of the method.

---

> ### Author Response · Authors · 2023-11-17
> **Response to Reviewer BzfQ**
>
> We thank the reviewer for the positive feedback and constructive suggestions.
>
> > W1. The intuition / reasoning behind the choice to prune the later layers to forget the spurious feature is not well-explained / discussed sufficiently.
>
> We have clarified the paragraph on "Intuition for lateTVG" at the end of Section 5.1 in the updated version. To summarize,
>
> 1. Existing SSL augmentations cause undesirable invariances in the representation space, which we demonstrate in the result that the spurious connectivity is greater than the invariant connectivity. This means that images with the same spurious feature are more likely to be mapped closer to each other in the representation space.
>
> 2. One approach would be to modify the augmentations in the image space, but here, we take the approach of augmenting the representation space.
>
> 3. Prior works [3, 4] have shown that, for a supervised network, spurious features are encoded in the final layers, and that perturbing the final layers is effective in removing the spurious feature. Furthermore, pruning has been shown to be more effective on minority examples [5].
>
> 4. If this intuition also holds for an SSL model, then the pruned network (which encodes invariant features more than spurious ones) provides a view of the samples in the representation space that do not include the spurious feature. In this case, the main encoder is forced to use the core feature to map representations to these views.
>
> > Q1. The problem of spurious correlations in SSL is actually equivalent to feature suppression.  It will be useful to discuss the equivalence of this problem as well as other relevant literature.
>
> We had cited the suggested work in the related work section when discussing representation learning under dataset imbalances and shortcuts (Chen et al., 2021). We are happy to cite any other suggestions for additional relevant prior work in a future revision.
>
> > Q2. In supervised learning, the models trained on Waterbirds etc. are initialized from ImageNet pretrained weights. Is that the case here as well?
>
> We use randomly initialized networks for our experiments, so spurious cues from ImageNet pretraining [e.g. 1] would not impact our evaluations. We explain this decision further in “The role of Initialization” paragraph at the end of Section 4.1.
>
> > Q3. Experiments on some larger scale datasets provided in packages such as WILDS or SpuCo can strengthen the empirical success of the method.
>
>
> We thank the reviewer for the suggestion. We found that the only dataset in WILDS with explicit spurious correlation is the CivilComments, which is a natural language dataset, and is outside the scope of this work. The other WILDS datasets do not appear to contain strong spurious cues to the best of our knowledge. SpuCo is certainly a relevant dataset, and we have cited it in our work. We hope to conduct experiments on this dataset for a potential camera-ready, but did not prioritize this as it includes subsets that are similar to our current datasets, such as waterbirds and CMNIST.
>
> Following the suggestion from Reviewer H1ZX, we have conducted experiments on Hard ImageNet [2], which have been added to Appendix F.1., and which we reproduce below:
>
>
> |              | None (&#8593;) | Gray (&#8595;) | Gray BBox (&#8595;) | Tile (&#8595;) |
> |--------------|----------------|----------------|--------------------|----------------|
> | SSL-Base     |   79.5       |     61.6        |        53.5         |       58.1      |
> | SSL-Late-TVG |      78.0        |    59.5          |        51.1          |     52.1           |
>
> We train an SSL model on the full Hard ImageNet train set and a linear classifier on the balanced subset (obtained by spurious rankings), then evaluate the classifier on the original test split and three additional splits with stronger color, shape, and texture spurious cues. This tests the model's ability to learn useful representations without relying on spurious correlations, which should result in lower accuracy on the manipulated splits, which we observe in the above table.
>
>
>
> [1] Meehan, Casey, et al. "Do SSL Models Have D\'eja Vu? A Case of Unintended Memorization in Self-supervised Learning." arXiv preprint arXiv:2304.13850 (2023)
> [2] Moayeri, Mazda, Sahil Singla, and Soheil Feizi. "Hard imagenet: Segmentations for objects with strong spurious cues." Advances in Neural Information Processing Systems 35 (2022): 10068-10077.
>
> [3] Lee, Yoonho, et al. "Surgical fine-tuning improves adaptation to distribution shifts." arXiv preprint arXiv:2210.11466 (2022).
>
> [4] Kirichenko, Polina, Pavel Izmailov, and Andrew Gordon Wilson. "Last layer re-training is sufficient for robustness to spurious correlations." arXiv preprint arXiv:2204.02937 (2022).
>
> [5] Hooker, Sara, et al. "What do compressed deep neural networks forget?." arXiv preprint arXiv:1911.05248 (2019).

---

> > ### Comment · Reviewer_BzfQ · 2023-11-19
> >
> > I thank the authors for their responses.
> >
> > I'd be eager to see the new experiments (that they've added for the rebuttal and those on SpuCo) in the final revision.
> >
> > I stand by my review to accept the paper.

---

### Official Review · Reviewer_H1ZX · 2023-11-01

**Soundness:** 3 good
**Presentation:** 3 good
**Contribution:** 3 good
**Rating:** 6
**Confidence:** 3

**Summary:**

The paper explores the impact of spurious correlations on Self Supervised Learning (SSL) for visual representation learning. The paper discusses how inductive biases in supervised learning favor simpler features, and when these features are spuriously correlated with the labels, it can result in suboptimal performance, especially for minority subgroups. The paper aims to investigate the extent to which SSL representations rely on spurious features for prediction.

The paper empirically demonstrates that common augmentations used in SSL can introduce undesired invariances in the image space, which may be problematic. It also finds that traditional approaches, such as dataset re-sampling during SSL, do not consistently lead to invariant representations. To address these findings, the paper proposes a new approach called LATETVG, which removes spurious information from SSL representations during pretraining by regularizing later layers of the encoder through pruning. LATETVG is shown to produce representations that outperform baselines on various benchmarks without requiring group or label information during SSL.

**Strengths:**

The strengths of the paper are as follows:

Theoretical Insights: By analyzing simpler cases (Section 3.3), the paper provides theoretical arguments that offer a deeper understanding of how common augmentations used in Self Supervised Learning (SSL) pre-training affect the model's reliance on spurious features for downstream linear classifiers.

Experimental evaluation of Spurious Feature Learning: The paper empirically explores the extent of spurious feature learning in self-supervised representations, focusing mainly on downstream worst-group performance. It demonstrates that traditional techniques for avoiding spurious correlations, such as re-sampling the training set with group information, do not consistently lead to improved core feature representations. This empirical analysis exposes the limitations of existing approaches and motivates the need for novel solutions.

LATETVG: The paper introduces LATETVG, a novel approach designed to correct biases introduced by augmentations. LATETVG modifies the views of samples in the representation space, effectively improving worst-group performance in downstream tasks on four datasets. This contribution presents a practical solution to the problem of spurious correlations in SSL pre-training, resulting in better core feature learning.

**Weaknesses:**

The paper is missing references to some important works on spurious feature learning:

1. Salient ImageNet: How to discover spurious features in deep learning?
2. WILDS: A Benchmark of in-the-Wild Distribution Shifts


Most of the results given in the paper are using smaller datasets. I believe the analysis of Section 4 could have been carried out a large number of publicly available SSL trained models. I would have liked to see some results using models trained on large datasets.

To evaluate the performance of models, the authors could have used more challenging datasets such as:
1. FOCUS: FOCUS: Familiar Objects in Common and Uncommon Settings
2. Hard ImageNet: Segmentations for objects with strong spurious cues
3. WILDS: A Benchmark of in-the-Wild Distribution Shifts

Most of the results seem to be on relatively simpler datasets.

**Questions:**

Why is the analysis of Section 4 limited to simpler datasets and no results are provided for the datasets discussed above?

---

> ### Author Response · Authors · 2023-11-17
> **Response to Reviewer H1ZX**
>
> Thank you for the review and for recognizing the theoretical insights, empirical analysis, and proposed method as strengths of our work.
>
> > W1. The paper is missing references to some important works on spurious feature learning
>
> Thanks for pointing out the missing references, we have cited these papers in the revised draft.
>
> > W2, Q1. Most of the results given in the paper are using smaller datasets. I believe the analysis of Section 4 could have been carried out a large number of publicly available SSL trained models. I would have liked to see some results using models trained on large datasets.
>
>
> We appreciate the suggestions of the three datasets. We note that WILDS does not contain strong spurious correlations except for the natural language dataset CivilComments. The image datasets in WILDS exhibit natural distribution shifts, but do not have the type of spurious correlations studied in our work to the best of our knowledge. FOCUS is definitely an interesting dataset to build upon, but for initial analysis on a larger scale dataset, it is unclear which attribute we should have chosen as the spurious attribute. We can add results on this dataset to a future version of this paper.
>
> Per your suggestion, we have added experiments on Hard ImageNet in Appendix F.1 to demonstrate our approach on a larger dataset. For the analysis in Section 4, we focused on datasets that contain the spurious attribute as a metadata, since our goal was to provide an in-depth empirical evaluation of spurious feature learning in SSL.
>
> We train an SSL model on the full Hard ImageNet train set and a linear classifier on the balanced subset (obtained by spurious rankings), then evaluate the classifier on the original test split and three additional splits with stronger spurious cues. This tests the model's ability to learn useful representations without relying on spurious correlations, which should result in lower accuracy on the manipulated splits.
>
>
> The summarized result table for Hard ImageNet is as follows:
>
>
> |              | None (&#8593;) | Gray (&#8595;) | Gray BBox (&#8595;) | Tile (&#8595;) |
> |--------------|----------------|----------------|--------------------|----------------|
> | SSL-Base     |   79.5       |     61.6        |        53.5         |       58.1      |
> | SSL-Late-TVG |      78.0        |    59.5          |        51.1          |     52.1           |
>
>
>
> As shown in this table, Late-TVG reduces accuracy on three Gray, Gray BBox, and Tile splits, meaning that the downstream classifier trained on Late-TVG representations relies less on the spurious (i.e. non-object) features. We have added more details and results on this experiment to Appendix F.1, and will include it in the revised main paper.
>
> We believe the added experiments on Hard ImageNet help address your concern about evaluation on larger datasets, and we hope to add the other large-scale dataset in a following version of this paper.

---

> > ### Author Response · Authors · 2023-11-21
> >
> > Dear Reviewer H1ZX,
> >
> > Thank you again for your valuable feedback. Since the author response period is ending in less than two days, we were wondering if our response has adequately addressed your concerns. If so, we would appreciate it if you could update your review and raise your score accordingly. We have added the suggested large-scale experiment to our paper. If there are any remaining questions or comments, please let us know and we would be happy to discuss.
> >
> > Thank you!

---

> > > ### Comment · Reviewer_H1ZX · 2023-12-04
> > > **Addressed my concerns**
> > >
> > > I would like to sincerely thank the authors for addressing my concerns. I am happy to raise the score.

---

### Official Review · Reviewer_GmKS · 2023-11-02

**Soundness:** 3 good
**Presentation:** 3 good
**Contribution:** 3 good
**Rating:** 6
**Confidence:** 3

**Summary:**

The paper investigates the impact of spurious features on Self-Supervised Learning (SSL) in visual representation learning. It starts by demonstrating that commonly used augmentations in SSL can inadvertently introduce undesired invariances in the image space, highlighting this issue with a simple example. The authors also examine classical approaches to mitigating spurious correlations, such as dataset re-sampling during SSL, and find these methods inconsistent in leading to invariant representations.

To address these challenges, the paper proposes a novel method called "LATE TVG," which aims to remove spurious information from representations during pretraining. This is achieved by regularizing later layers of the encoder via pruning. The authors present empirical evidence showing that LATE TVG produces representations that outperform baselines on several benchmarks. Notably, their method does not require group or label information during SSL, marking a significant advantage over traditional approaches.

The paper provides a thorough examination of the pitfalls of inductive biases in supervised learning, particularly when dealing with minority subgroups and spurious correlations. The proposed solution, LATE TVG, is a promising step towards more robust and fair SSL models, showing effectiveness in various benchmarks without relying on group or label information.

**Strengths:**

**1. Originality:**
The paper introduces a novel method, "LATE TVG," to mitigate the influence of spurious correlations in Self-Supervised Learning (SSL) for visual representation learning. This approach is original in its utilization of later layer regularization via pruning to enhance the robustness of SSL models. Unlike conventional methods that often rely on re-sampling or require group or label information, LATE TVG innovatively ensures invariant representations without such dependencies. The idea of addressing undesired invariances introduced by common augmentations in SSL is a creative combination of existing concepts in a unique problem formulation.

**2. Quality:** The empirical evidence presented in the paper is of high quality, showcasing the effectiveness of LATE TVG through comprehensive benchmarks. The authors provide a detailed analysis of the pitfalls of standard augmentations in SSL and demonstrate the superiority of their method over traditional approaches. The experiments are well-designed and executed, offering convincing support for the proposed solution. The quality of the research is further underscored by the rigorous testing on various benchmarks, which highlights the method's robustness and generalizability.

**3. Clarity and Significance:** The paper is clearly written, with a well-structured format that guides the reader through the problem statement, methodology, and findings. The significance of the work is evident, as it addresses a critical challenge in SSL—ensuring fairness and robustness in visual representation learning. By providing a solution that does not require group or label information during SSL, the paper makes a useful contribution to the field, potentially leading to more equitable and effective machine learning models.

**Weaknesses:**

**1. Scalability and Efficiency:** The paper introduces the LATE TVG method, which involves regularizing later layers of the encoder via pruning. While this approach is novel, the scalability and computational efficiency of the method in large-scale settings are not thoroughly addressed. Pruning, especially in deeper layers, can be computationally intensive and may not scale well with very deep networks or extremely large datasets.

**Actionable Insight:**
Future work could focus on optimizing the pruning process to enhance scalability and efficiency. This might involve investigating more computationally efficient pruning techniques or adapting the method to work with sparse neural networks.

**2. Domain Generalization:** Although the paper demonstrates the effectiveness of LATE TVG across several benchmarks, it primarily focuses on visual representation learning. The generalizability of the method to other domains or types of data, such as text or audio, is not explored.

**Actionable Insight:**
To strengthen the paper's contributions, additional experiments could be conducted to evaluate the method's performance on non-visual datasets. This would provide a clearer understanding of the method's applicability across different domains and data modalities.

**3. Comparison with State-of-the-Art:** While the paper shows that LATE TVG outperforms traditional methods, it does not provide an extensive comparison with the latest state-of-the-art methods in SSL that address similar challenges. Without this context, it's difficult to gauge the relative progress made.

**Actionable Insight:**
The authors could enhance the paper by including a more comprehensive comparison with state-of-the-art methods. This might involve reproducing results from recent papers or directly comparing the proposed method against other cutting-edge techniques in SSL.

**Questions:**

Here are some questions and suggestions for the authors:

**1. Clarification on Scalability and Efficiency:**
- Question: Could you provide more details on the computational efficiency and scalability of the LATE TVG method, especially when applied to very large datasets or extremely deep networks?
- Suggestion: It would be beneficial if the authors could include a section discussing the computational complexity of their method, possibly comparing it with other SSL methods in terms of training time and resource utilization.

**2. Long-Term Effects of Pruning:**

- Question: Pruning, especially in later layers, might have long-term effects on the learning capabilities of the network. Have you investigated how the pruning process affects the network's ability to learn new tasks or adapt to new data over time?
- Suggestion: Providing insights or conducting experiments on the long-term effects of pruning could offer a more nuanced understanding of the method's robustness and adaptability.

**3. Robustness to Adversarial Attacks:**

- Question: How robust is the LATE TVG method to adversarial attacks, given that it focuses on mitigating spurious correlations?
- Suggestion: Including experiments or discussions on the method's robustness to adversarial examples could highlight another dimension of its efficacy, particularly in security-sensitive applications.

---

> ### Author Response · Authors · 2023-11-17
> **Response to Reviewer GmKS**
>
> Thank you for your positive feedback on our work and the suggestions for improvement. We appreciate that you recognize the originality, quality, and significance of our work.
>
> > W1, Q1. Scalability and Efficiency
>
> The SSL-LateTVG model updates the same number of parameters as SSL-Base during training, with the forward pass keeping both the original and pruned encoder (See Algorithm 1 in Appendix B). The pruning operation is cost O(n) where n is the number of parameters. So any FLOPs used for the extra pruning mechanism will be very small compared to a single forward pass.
>
> In summary, the training parameters are identical between SSL-Base and LateTVG, with the LateTVG encoder modified via pruning. For the backward pass, only the original encoder and other SSL parameters like the projection head are updated. We have also clarified this computational process in Appendix B, and parameters in Appendix D.2.
>
> We provide runtimes on the MetaShift dataset below as an example:
>
> SimSiam: SSL-Base: 4h 26m, SSL-LateTVG: 5h 14m
>
> SimCLR: SSL-Base: 4h 39m SSL-LateTVG: 4h 50m
>
>
> > W2. Domain Generalization
>
> We appreciate the reviewer raising this point. We focus on the image modality in this work because SSL methods we use as base implementations are designed for images. We agree that evaluating our approach on non-visual modalities is an important direction for future work.
>
> > W3. Comparison with State-of-the-Art
>
> To the best of our knowledge, this is the first work evaluating visual self-supervised representation learning on datasets containing known spurious correlations. If the reviewer could suggest recent SSL papers addressing similar challenges, we would be happy to include comparisons.
>
>
> > Q2. Long-Term Effects of Pruning
>
> While we agree that this is an interesting direction, our focus is on mitigating spurious correlations for self-supervised representation learning itself. We leave investigations into pruning's effects on adapting to new data or tasks to future work.
>
>
> > Q3. Robustness to Adversarial Attacks
>
> We believe that evaluating adversarial robustness is orthogonal to the problem studied in this paper. Our method targets removing spurious correlations, not increasing robustness to adversarial examples. While an interesting direction, analyzing LateTVG's robustness to adversarial attacks is out of scope for this work. We leave this to future work and maintain our focus on addressing spurious correlations in self-supervised learning.

---

> > ### Author Response · Authors · 2023-11-21
> >
> > Dear Reviewer GmKS,
> >
> > Thank you again for your valuable feedback. Since the author response period is ending in less than two days, we were wondering if our response has adequately addressed your concerns. If so, we would appreciate it if you could update your review and raise your score accordingly. We have answered your questions regarding weaknesses mentioned, and have added details related to efficiency to the paper. If there are any remaining questions or comments, please let us know and we would be happy to discuss.
> >
> > Thank you!

---

### Author Response · Authors · 2023-11-17
**General Author Response**

We would like to thank the reviewers for their thoughtful comments and valuable feedback. We have responded to the comments and made changes to our submission. In particular:

1. Added experiments on Hard ImageNet dataset to Appendix F.1 to demonstrate results on larger datasets (In response to reviewer H1ZX, and BzfQ)

2. Included Late-TVG algorithm details in Appendix B to improve comprehension. (In response to reviewer 32bS)

3. Clarified notation and overload of symbols in the theory section to improve presentation, and added details on how the connectivity error rates are computed to the main paper. (In response to reviewer 32bS)

We are also grateful that the reviewers recognized the following strengths of our work:

1. We depict the prevalent issues arising in SSL from reliance on spurious features, providing a cogent problem formulation (Reviewer 32bS).

2. Our theoretical analysis offers valuable insights into how augmentations affect dependence on spurious cues in SSL (Reviewer H1ZX).

3. This is the first study analyzing the significant problem of spurious correlations in SSL, considering the lack of explicit group information (Reviewer BzfQ). Through thorough empirical analysis, we show limitations of existing correlation mitigation approaches (Reviewer H1ZX, BzfQ).

4. Our introduced method successfully reduces spurious correlations and improves downstream performance (Reviewers 32bS, BzfQ, GmKS).


5. The originality of our novel method and problem formulation: addressing spurious correlations in SSL without relying on group labels (Reviewer GmKS).

We have provided detailed responses to the questions and concerns brought up in your review. Please let us know if any clarification is required or reach out with additional questions you may have so that we can improve the work through this productive review process.

---

### Meta-Review · Area_Chair_fJM1 · 2023-12-12

**Metareview:**

This paper studies spurious correlations in self-supervised learning (SSL) in computer vision. The paper shows that commonly used augmentations in SSL can cause undesired invariances, and uses a toy example to illustrate this. Then the paper proposes a novel algorithm LateTVG to alleviate this issue by pruning the later layers during pre-training.

This paper is well-motivated, clearly written, and has solid contributions both to the fundamental understanding of spurious learning in SSL and to practical improvements of this issue via a novel algorithm. The questions from the initial reviews were well addressed.

**Justification For Why Not Higher Score:**

The reviews had some reservations about the applicability of the LateTVG algorithm and wanted to see results in more challenging datasets and other domains.

**Justification For Why Not Lower Score:**

The paper has made solid contributions and has no significant weekness.

---

### Decision · Program_Chairs · 2024-01-16

Accept (spotlight)